# Evaluation of Five Mammalian Models for Human Disease Research Using Genomic and Bioinformatic Approaches

**DOI:** 10.3390/biomedicines11082197

**Published:** 2023-08-04

**Authors:** Sankarasubramanian Jagadesan, Pinaki Mondal, Mark A. Carlson, Chittibabu Guda

**Affiliations:** 1Department of Genetics, Cell Biology and Anatomy, University of Nebraska Medical Center, Omaha, NE 68198, USA; s.jagadesan@unmc.edu (S.J.); macarlso@unmc.edu (M.A.C.); 2Department of Surgery and Center for Advanced Surgical Technology, University of Nebraska Medical Center, Omaha, NE 68198, USA; pinaki.mondal@unmc.edu; 3Center for Biomedical Informatics Research and Innovation, University of Nebraska Medical Center, Omaha, NE 68198, USA

**Keywords:** animal models, NHPs, rodents, marmoset, sequence similarity, SNPs, human diseases

## Abstract

The suitability of an animal model for use in studying human diseases relies heavily on the similarities between the two species at the genetic, epigenetic, and metabolic levels. However, there is a lack of consistent data from different animal models at each level to evaluate this suitability. With the availability of genome sequences for many mammalian species, it is now possible to compare animal models based on genomic similarities. Herein, we compare the coding sequences (CDSs) of five mammalian models, including rhesus macaque, marmoset, pig, mouse, and rat models, with human coding sequences. We identified 10,316 conserved CDSs across the five organisms and the human genome based on sequence similarity. Mapping the human-disease-associated single-nucleotide polymorphisms (SNPs) from these conserved CDSs in each species has identified species-specific associations with various human diseases. While associations with a disease such as colon cancer were prevalent in multiple model species, the rhesus macaque showed the most model-specific human disease associations. Based on the percentage of disease-associated SNP-containing genes, marmoset models are well suited to study many human ailments, including behavioral and cardiovascular diseases. This study demonstrates a genomic similarity evaluation of five animal models against human CDSs that could help investigators select a suitable animal model for studying their target disease.

## 1. Introduction

A comprehensive understanding of the genomic relatedness between humans and other mammals is necessary to evaluate the common pathways and functional similarities and the appropriateness of using different animal models to study human diseases. Although the mouse is the most commonly used animal model in the study of human diseases [1], its smaller size and lifespan and differences in the latency periods for diseases [2], drug metabolism [3], inflammatory response [4], and other processes [5,6,7,8] suggest the need to identify animal models with more disease-relevant similarities to humans for the study of human diseases. A comparison of genomic sequences among human, mouse, and pig genomes indicated that pig sequences were closer to the human sequences, with greater numbers of ultra-conserved regions compared to the mouse genome [9,10,11]. Nonhuman primates are evolutionarily more closely related to humans. After the ban on working with chimpanzees and other great apes, macaques have become the most closely related nonhuman primate model used to study human diseases [12]. Human disease genes and known drug domains have shown high degrees of similarity with the rhesus macaque genome [13]. The marmoset, on the other hand, makes for a useful animal model due to its short gestation and sexual maturation periods and its greater sequence similarity with humans compared to rodents [12]. It is also a useful model for studying diseases in neurobiology [14]. A comparative analysis of disease-associated genetic variations across commonly used animal models will deepen our understanding of the similarities, leading to the appropriate use of specific animal models for disease-specific research.

In this study, we retrieved protein-coding sequences (CDS) from two rodents (a mouse and a rat), a pig, two nonhuman primates (a rhesus macaque and a marmoset), and humans to identify conserved CDS across the six species. A multiple sequence alignment was performed across the six species using the conserved CDS, and for each species, mapped positions corresponding to human single-nucleotide polymorphisms (SNPs) were extracted from the alignment. The mapped SNPs were queried for disease associations to identify common (human SNPs that are identified in all other species) and species-specific clinically associated SNPs to better define the relevance of an animal model to the study of various human diseases. Taken together, the genome comparison performed in this study will provide some insights into selecting suitable animal models for studying human diseases.

## 2. Materials and Methods

### 2.1. Retrieval of Protein Coding Sequences

Human, rhesus macaque, pig, mouse, and rat genomic data were retrieved from Ensembl [15], and marmoset genomic data were retrieved from the National Center for Biotechnology Information (NCBI). The datasets included *Homo sapiens* (human—GCA_000001405.28), *Macaca mulatta* (rhesus macaque—GCA_003339765.3), *Callithrix jacchus* (marmoset—NCBI: GCF_009663435.1), *Sus scrofa* (pig—GCA_000003025.6), *Mus musculus* (mouse—GCA_000001635.9), and *Rattus norvegicus* (rat—GCA_000001895.4). The downloaded genomes were assembled at the chromosomal level, with the scaffold N50 ranging from ~14 to 106 million base pairs, which showed good assembly quality. The genome assembly information, such as the genome length, size, the scaffold and contig of N50 and L50, and assembly level, is provided in Appendix A. We extracted a total of 19,962 human, 21,591 rhesus macaque, 22,252 marmoset, 21,280 pig, 21,848 mouse, and 22,250 rat coding sequences (CDSs) for the current analysis. A CDS is a DNA sequence that represents all the protein-coding exons concatenated into one continuous sequence.

### 2.2. Identification of Similarities between Human CDSs and Other Mammalian Sequences

Five pairwise sequence comparisons that included human vs. rhesus macaque, human vs. marmoset, human vs. pig, human vs. mouse, and human vs. rat were performed to identify human CDSs that are conserved across the animal models. The Basic Local Alignment Search Tool (BLAST) database was constructed for the human CDS set using the makeblastdb application. The blastn tool was used to align each non-human query CDS against each human CDS using the following algorithmic options: -max_hsps 1 -max_target_seqs 1 in BLAST+ (version 2.7.1) [16]. Based on the pairwise alignment, we identified conserved CDSs in other mammalian models against the human model, in which a conserved sequence is defined as a single contiguous sequence from each species that also passes the following filters: (i) It shares at least 50% identity with the human CDS and (ii) it covers at least 50% of the length of the human CDS. Alignments that failed to meet either of these criteria were excluded from further analysis. Based on these similarities, conserved sequences were identified across the five comparisons and plotted using UpSetR [17]. To understand the synteny block distribution for the human conserved CDSs on different chromosomes of the five species, we created circos plots using the R package shinyCircus [18], using the human chromosome as the reference. A one-way ANOVA and Bonferroni’s multiple comparisons test were performed using GraphPad Prism 10 (www.graphpad.com, accessed on 28 March 2022, GraphPad Software (version 10), Boston, MA, USA) to calculate the statistical significance of the percent identities of the CDS.

### 2.3. Comparison of Conserved CDS and the Identification of SNPs and Their Associated Diseases

A multiple sequence alignment was performed across the six species, using 10,316 conserved CDSs via ClustalW2 [19], and human SNPs were extracted from the alignment using SNP-sites [20] and msa2snp (https://github.com/pinbo/msa2snp, accessed on 5 April 2022). The Ensembl Variant Effect Predictor [21] was used to identify the SNPs with rsID (RefSeq). Later, Ensembl Post GWAS and SNPnexus (which use Cosmic, ClinVar, and GWAS) [22] were used to identify SNPs associated with human diseases. It should be noted that the major allele, as well as any minor allele, may serve as a disease allele depending on their penetrance levels and other covariates, which are not specifically analyzed in this study. These diseases were classified into 24 different categories using DisGeNET [23]. Further, we determined the extent of the disease associations of the SNPs for each species as the percentage of the number of diseases in each category divided by the total number of diseases associated with all SNPs in that species. We also identified the conserved (human SNPs that are identified in all other species) and species-specific SNPs (human SNPs that are identified only in a given species) associated with diseases for a particular animal model based on the major allele match in the human SNP. These SNPs were plotted using R packages, such as shinyCircus [18] and karyoploteR [24].

### 2.4. Construction of a Phylogenetic Tree

Using EMBOSS Union, multiple sequence alignments of 10,316 conserved CDSs from six organisms were concatenated as per their order on the human chromosome (1-22, X, and Y) [25]. The phylogeny was constructed using FastTree (parameter –nt –gtr), version 2.1 [26], and visualized using Molecular Evolutionary Genetics Analysis (MEGA), version 11 [27].

## 3. Results

### 3.1. Identification of Conserved CDSs with Human Sequences

We compared the coding sequences between the human genome and five other species using the BLAST program, with cutoffs of a sequence identification of at least 50% and a length match of 50% to the human sequences. A detailed workflow for the analysis is provided in Figure 1. The results showed that the rhesus macaque has the highest average identity (96.82%), followed by the marmoset (94.65%), pig (89.37%), mouse (86.65%), and rat (86.53%) (Table 1). The percent identity ranged from 100 to around 70, which is comparable across all comparison groups (Table 1). However, the distribution of the percent identity of the CDSs is not uniform in all comparison groups. In the rhesus macaque and marmoset, the identity distribution is skewed towards the median (the median for the rhesus macaque is 97.29, and for the marmoset, the median is 95.29, Appendix A), denoting that the majority of the CDSs in these primate species are highly identical to human CDSs (Figure 2a). On the other hand, the identities in the pig, mouse, and rat are more widely distributed around the median, suggesting a varying degree of similarity with certain gene families of the human genome (Figure 2a and Appendix A). Among these three organisms, pigs showed the highest median value (89.89% identity) and a significantly higher percent identity with human CDS than mice or rats (Figure 2a). The complete result of this identity analysis is provided in Appendix A.

Based on the pairwise alignment of CDSs between the human genome and the five other species, 10,316 CDSs were found to be conserved across all six species (Figure 2b and Appendix A); these were used for further analyses. In all species, this set of conserved CDSs recorded higher percentage identities than those involving all CDSs (Table 1). Among the non-human primates, the rhesus macaque showed the highest average percentage identity with the human genome at 97.53%, and the pig demonstrated a percentage identity of 90.38%, which is significantly higher than the percentage identities of mice and rats (Figure 2c and Appendix A).

Next, we mapped all the conserved CDSs from each non-human species to matching positions (determined via similarity) on human chromosomes to understand their synteny distribution in each species. For visualization purposes, we used a common color-coding scheme for each chromosome number, in which the same color represents the same chromosome number in all species. Notably, marmosets have the same number of chromosomes as humans, but the other species have fewer. The pig has the lowest number, with only 18 chromosomes. The circos plots (Figure 3a) illustrate the mapping of synteny blocks (represented by CDSs) from different chromosomes of the non-human species, using the human chromosomal numbers as a reference. The chromosomes in the non-human species mapped with only one color indicate that they contain corresponding intact human synteny blocks, and those shown with mosaic coloring indicate that the human synteny blocks are distributed on different chromosomes, as indicated by different colors. For instance, the synteny blocks of conserved CDSs from chromosome 1 of the macaque (shown in red) also map to human chromosome 1, but the corresponding synteny blocks from other species are mapped to different human chromosomal locations. Similarly, the synteny blocks from chromosomes 12 and 13 of the macaque are mapped to human chromosome 2. Notably, the synteny blocks on chromosomes 17, 20, and X are intact in a single chromosome in all species (as indicated by only one color), while those from other chromosomes are fragmented and distributed in multiple chromosomes (shown with mosaic color mapping) (Table 2).

A phylogenetic analysis based on the conserved CDS examined the evolutionary distances among the six species. As shown in Figure 3b, the nonhuman primate (NHP) group has the closest distance to the human genome, with the pig positioned in the middle and the rodent group being the farthest from the human genome. The chromosome-specific mapping for all the CDSs and conserved CDSs is presented in Appendix A, respectively.

### 3.2. Mapping Human Disease-Relevant SNPs in Other Species

The mapping of human SNPs to the other species after the multiple sequence alignment of 10,316 conserved CDSs showed differences in the SNP numbers between the primates and the other three species. The primates had a higher number of predicted SNPs (reference SNPs from the dbSNP database with matching nucleotides for the human major alleles in other genomes; rhesus macaque, 577,417, and marmoset, 516,545) in the conserved CDSs than the pig (395,787), mouse (264,070), and rat (256,017) genomes, as anticipated. A full list of all the mapped SNPs is provided in Appendix A. The SNPs were then annotated for disease association and plotted on each chromosome to easily visualize the distribution and variation of the SNPs across the species (Appendix A). The identified diseases with their corresponding rs IDs for the human versus five animal genomes are listed in Appendix A. Among the predicted diseases based on the human SNPs, 1082 were conserved among all six species (Figure 4a). The predicted diseases were then classified into certain disease categories and compared amongst the five non-human model organisms based on their relative percentage in each species. The higher the percentage, the more relevant the model is to the study of human disease. SNPs in cancer and congenital, hereditary, and neonatal diseases and abnormalities and nervous system diseases were highly prevalent in all the models (Figure 4b), while some disease classes were specific to a model organism. SNPs associated with musculoskeletal diseases were specifically observed in the rhesus macaque, and SNPs associated with behavioral and cardiovascular diseases were observed in the marmoset. The relative percentage of diseases associated with cancer, gastrointestinal diseases, and organismal injury and abnormalities was higher in pigs, while a higher number of diseases associated with nutritional and metabolic diseases was found in mice (Figure 4b). These results indicate that human SNPs associated with specific disease classes are prevalent in specific model species, which may provide a basis for the selection of an appropriate model for a specific disease.

The number of disease associations varies between species. The rhesus macaque had the highest number (53) of diseases associated with mapped SNPs in 42 genes, while the marmoset and pig showed higher numbers (24 diseases with 20 mapped genes in the marmoset and 23 diseases with 19 genes in the pig) than the mouse (7 diseases and 5 genes) and rat (8 diseases and 6 genes) (Table 3). We provide a curated list of all the model-specific human-associated diseases and identified SNPs with their corresponding genes in Appendix A.

Next, we identified the human SNP-bearing genes in the animal models that showed an association with a human disease, as shown in the color-coded chromosomal map (Figure 5). For each animal model, the genes that showed the highest number of disease associations corresponding to each human chromosome are listed in Appendix A. The full list of genes per chromosome is provided in Appendix A. Ten common genes (MUTYH, MSH6, APC, DSP, RET, POLE, RB1, FBN1, TSC2, and FLNA) were identified across all species. In our study, none of the disorders were associated with the mapped SNPs on chromosome Y. The rhesus macaque, marmoset, and pig have 17 common disease-associated genes among them, while the mouse and rat have 16 common genes, indicating two separate groups of model organisms in terms of the mapped SNPs in the conserved CDSs and disease associations. The SNPs identified in the APC gene have the highest number of disease associations across species.

## 4. Discussion

The selection of an animal model for studying a human disease is partly based on the animal’s genetic relatedness to humans. However, species evolution is generally not synchronous with gene evolution, which results in certain human genes having more similarity with genes in certain model organisms; this can influence the selection of different animal models for different research projects. Although primates are known to be evolutionarily closer to humans and can better mimic human physiology, the use of NHPs is expensive, time consuming, heavily regulated, and subject to availability [28,29]. In this study, we identified genetic similarities between CDSs and mapped disease-associated human SNPs with commonly used animal models, including the rhesus macaque, marmoset, pig, mouse, and rat.

The identification of NHPs as humans’ closest evolutionary neighbors was expected. The BLAST-based genome-wide sequence comparison of the rhesus macaque and marmoset genomes with the human genome showed a 95–97% similarity across about 18,000 sequences, and the CDS-level identities also registered a similarity of 96–98% in these NHP models, further supporting the belief that the macaque and marmoset are good animal models to use in the study of human diseases (Table 1). Alternatively, the pig’s genome-level similarity (89.4%) with the human genome was higher than that of the rat (86.7%) or mouse (86.5%). This observation remained true at the CDS level, suggesting that pigs make a more suitable model (with respect to genomics) for the study of human diseases than rodents (Table 1).

Other than the nonhuman primates, pigs share the highest number of common genes containing disease-associated SNPs with humans when compared to the other examined species (Table 3), favoring the pig model for the study of genetically predisposed human diseases. Similarly, these common SNPs between pigs and humans are also associated with the highest number of human diseases, further emphasizing the pig’s strong disease relatedness to humans. Although mice and rats are phylogenetically closer to humans than pigs [10], possible reasons for finding more SNPs that mapped to the pig genome compared to a rodent genome could be due to (i) the shortened generation time in rodents, which leads to expedited divergence, and (ii) the coverage of only a subset (10,316 conserved CDSs) of the total genome that did not account for all the variations in the whole genome. We mapped the SNP-associated diseases onto different disease groups and observed that the relative frequency of diseases in the cancer category was the highest in the pig; however, we were not able to test these differences statistically because SNP mapping had been carried out against a single reference genome for each species, resulting in a lack of a distribution of values for performing statistical tests. Nevertheless, an examination of multiple levels of relatedness between humans and pigs (with respect to genomes, CDSs, SNPs, and cancer disease levels) suggests that the pig would be a more accurate genetic model for human cancer research than rodents. It should be noted that the observations and inferences made in this study are limited only to the disease-associated SNPs in the CDS regions of the genome, while some disease-associated SNPs also exist in the intronic and other regions of the genome, which were not included in this study.

SNP-associated behavioral diseases were mostly observed in the marmoset, which is in concordance with the existing literature [30,31]. Our analysis also suggests that the marmoset could also be used to model cardiovascular diseases, which are found in marmosets in captivity [32,33]. Musculoskeletal diseases were found to be frequent in rhesus macaques, which undergo structural changes to their musculoskeletal biology that contribute to their increasing frailty with age [34]. On the other hand, the rodent models showed higher numbers of SNPs associated with the reproductive system and metabolic diseases. Rodents, specifically rats, have been used to study reproductive diseases [35]. The mouse has been widely used to study human metabolic diseases [36].

When we specifically analyzed the chromosomes and genes that are associated with human diseases, the genes involved in the DNA repair pathway were the most commonly found genes in all the species. PTCH1 was the topmost SNP-prevalent gene on chromosome 9 in pigs. PTCH1 was found to be altered in 2.76% of all cancers but was predominantly altered in colon cancer (TCGA data portal, My Cancer Genome database [37]). Similarly, in the mouse, STK11, which was most commonly altered in lung cancer, appeared as the topmost gene on chromosome 19. STK11 was targeted to create a pig model of lung cancer that shows evidence of inflammation; however, more work is required to fully develop the model [38].

## 5. Conclusions

By using genome-wide coding sequence similarities and mapping human SNPs onto the genomes of five other mammalian species, this study demonstrated important similarities and differences among major model organisms with respect to humans. Based on the sequence analysis, some species (e.g., NHPs) appeared to make superior models for the study of human diseases compared to other species. Overall, it was determined that the pig has a greater degree of sequence homology with humans than rodents. Based on these data, the pig emerges as a reasonable model for use in studying human diseases, most notably cancer, in which a related immune system is present. Marmoset models are well positioned to study behavioral and cardiovascular diseases. Rodent models could be better for studying the reproductive system and metabolic diseases, but they have an obvious size discrepancy and less overall sequence homology with humans than pigs. This study presents a suitability assessment based on the available genomic data only; however, other factors, such as cost, feasibility, and individual project goals, should be carefully considered in selecting an appropriate animal model for each research project.

## Figures and Tables

**Figure 1 biomedicines-11-02197-f001:**
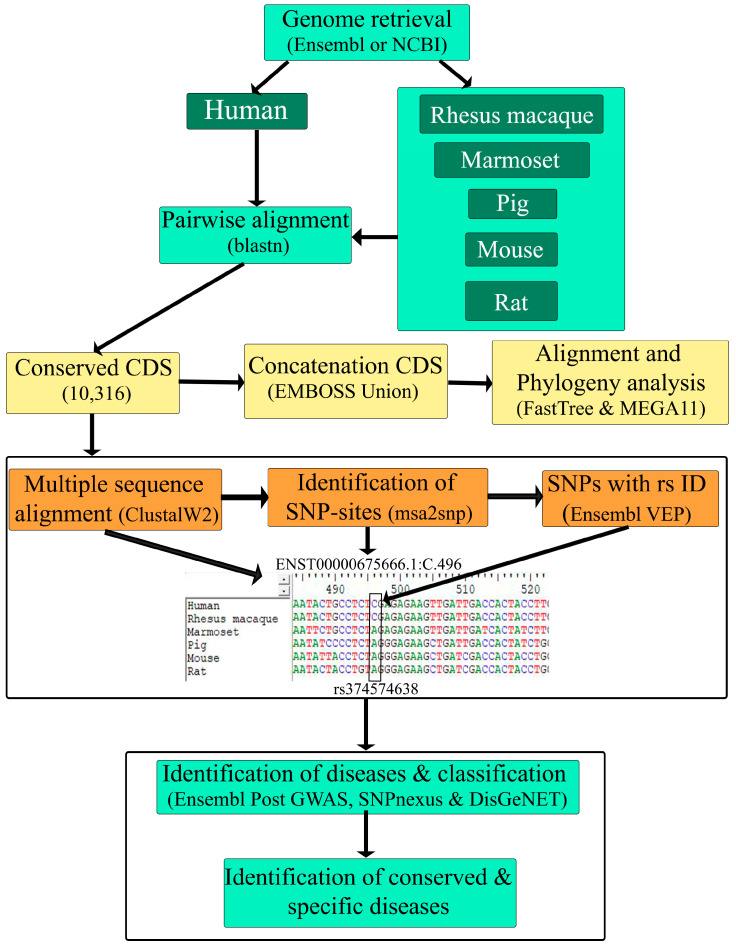
Detailed workflow of CDS-based comparison across species and the identification of conserved and specific diseases.

**Figure 2 biomedicines-11-02197-f002:**
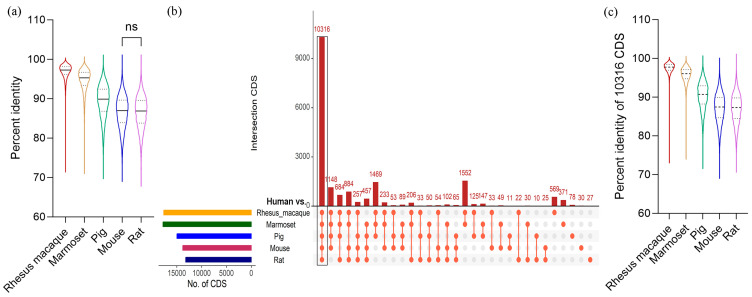
Similarity of coding sequences (CDSs) between human genome and animal models. (**a**) Distribution of the CDS identity percentages between the human genome and five different animal models. The line inside the violin plot represents the median values. (**b**) The total number of mapped CDSs in five different species against the human genome. The upset plot shows intersections across the five comparisons. Each bar represents the number of mapped CDSs, and the orange dot below the bar indicates their conservation status across each species. (**c**) Distribution of percentage identities of 10,316 conserved CDSs between the human genome and five animal models. The line inside the violin plot represents median values. Comparisons of the percent identities between the species are statistically significant unless noted by ns = statistically non-significant, as determined via a one-way ANOVA and Bonferroni’s multiple comparison test (*p*-values are provided in Appendix A).

**Figure 3 biomedicines-11-02197-f003:**
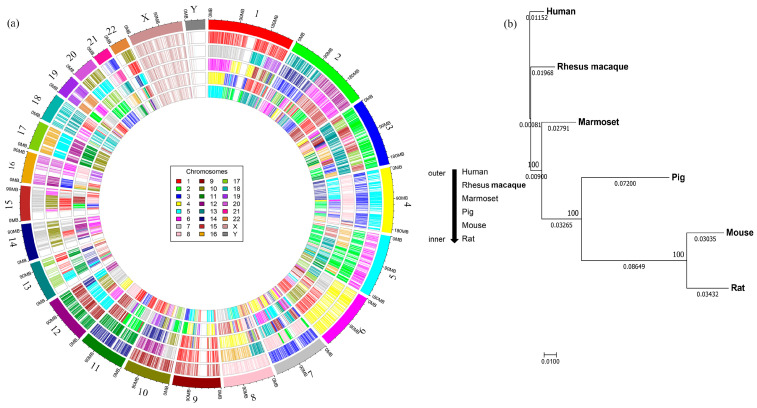
Mapping of the synteny blocks of model organism chromosomes onto human chromosomes and phylogenetic analysis. (**a**) The Circos plot shows the positions of 10,316 conserved CDSs for five different animal models, using a unique color for each chromosome number. The outermost circle represents the color-coded human chromosomes, and each inner circle represents the mapping of the synteny blocks of chromosomes from each species onto the human chromosomes, showing how they are distributed across the human chromosomes. The number of chromosomes varies across each species; rhesus macaques have Chr1-20, X, Y; marmosets have Chr1-22, X, Y; pigs have Chr1-18, X, Y; mice have Chr1-21, X, Y; and rats have Chr1-20, X, Y. (**b**) A phylogenetic tree was constructed based on the 10,316 conserved CDSs, which showed that the evolutionary distance from the human genome was the shortest for the rhesus macaque, followed by the marmoset, pig, mouse, and rat. The values above and below the line indicate the bootstrap numbers and the evolutionary distances between the species.

**Figure 4 biomedicines-11-02197-f004:**
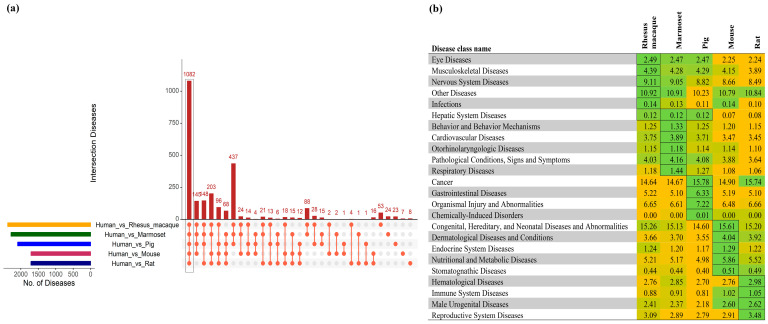
Comparison of the SNP-associated human diseases across the animal models. (**a**) An upset plot shows the intersections of the SNP-associated human diseases across the five species. Each bar represents the number of identified diseases, and the orange dot below the bar indicates their conservation across the comparisons. (**b**) The diseases were classified into 24 different categories, and the percentages of a specific disease class in each animal model were plotted. Higher to lower percentage numbers within species are colored from green to yellow. The highest value across species for each disease class is indicated by a box.

**Figure 5 biomedicines-11-02197-f005:**
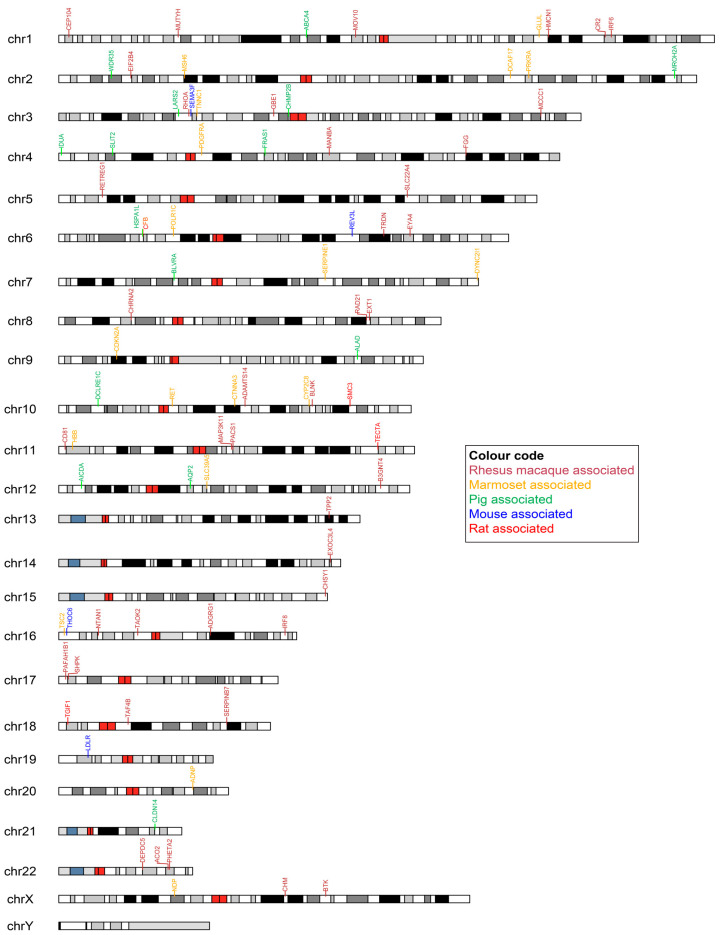
Mapping of human SNP-associated genes in different animal models across human chromosomes. The identified SNP-associated genes for species-specific diseases are represented with different colors using Karyoplot.

**Table 1 biomedicines-11-02197-t001:** Comparison of sequence identity among the CDSs between the human genome and five other mammalian animal models.

Comparison	Identified BLAST Hits *	Average Percentage Identity	Range of Percent Identity	Average Percentage Identity for Conserved CDS
Human vs. rhesus macaque	17,638	96.82	100–71.74	97.53
Human vs. marmoset	17,787	94.65	100–71.63	95.76
Human vs. pig	14,992	89.37	100–70.81	90.38
Human vs. mouse	13,806	86.65	100–70.11	87.19
Human vs. rat	13,222	86.53	100–68.93	87.04

* A sequence identiy of at least 50% and a 50% or higher length match to the human CDS.

**Table 2 biomedicines-11-02197-t002:** Chromosome-specific mapping of conserved CDSs between human and animal model genomes.

Human Chromosomes	Total CDS	Conserved CDS	Rhesus Macaque *	Marmoset *	Pig *	Mouse *	Rat *
Chr1	2049	1088	1	7, 18, 19	6, 4, 9, 10, 14, 2, 7	4, 3, 1, 8	5, 2, 13,19, 14, 10, 17, 4
Chr2	1244	750	12, 13	6, 14	15, 3	1, 2, 6, 17, 12, 11	9, 6, 3, 4, 14, 13, 20, 18
Chr3	1075	645	2	15, 17	13	9, 16, 3, 6, 14	8, 11, 2, 4, 16, 15
Chr4	752	390	5	3	8, 15, 14	5, 3, 8	14, 2, 16, 19, 4
Chr5	883	502	6	2	2, 16	13, 18, 11, 15	2, 18, 10, 17, 1, 9
Chr6	1045	574	4	4	7, 1	17, 10, 13, 9, 4, 1	20, 1, 17, 9, 8, 5
Chr7	919	470	3	8,2	18, 9, 3	5, 6, 12, 11, 13	4, 12, 6, 14, 17
Chr8	684	372	8	16, 13	4, 14, 17, 15	15, 8, 14, 4, 1, 3	7, 5, 16, 15, 2, 11
Chr9	779	402	15	1	1, 10, 14 3	4, 2, 19, 13	5, 3, 1, 17
Chr10	1309	619	9	12, 7	14, 10	19, 14, 2, 10, 7, 18, 6, 13	1, 17, 20, 16, 15, 4
Chr11	727	432	14	11	2, 9	7, 9, 19, 2	1, 8, 3
Chr12	1033	582	11	9	5, 14	10, 5, 6, 15	7, 12, 4
Chr13	321	182	17	1, 5	11	14, 8, 5, 3	15, 16, 12, 2, 9
Chr14	610	360	7	10	7, 1	12, 14	6, 15
Chr15	596	371	7	10, 6	1, 7	9, 2, 7	8, 3, 1
Chr16	851	378	20	12, 20	6, 3	8, 7, 16, 17, 11	19, 1, 10
Chr17	1182	637	16	5	12	11	10
Chr18	269	157	18	13	1, 6	18, 17, 1	18, 9, 3
Chr19	546	282	19	22	6, 2	7, 8, 10, 17, 9	1, 7, 16, 8, 19, 9, 12
Chr20	1469	457	10	5	17	2	3
Chr21	234	76	3	21	13	16, 10, 17	11, 20
Chr22	444	202	10	1	5, 14	15, 11, 16, 5, 10	7, 14, 11, 12, 20
ChrX	853	381	X	X	X	X	X
ChrY	46	7	Y	Y, X	Y, X	Y, X	Y, X

* The chromosome numbers are listed according to the highest to lowest number of CDSs mapped to the human genome.

**Table 3 biomedicines-11-02197-t003:** Number of human SNPs mapped to the conserved CDSs across five animal models and information about their associated diseases.

Organisms	Total SNPs in 10,316 CDSs with RS Number	SNPs Associated with Disease	No. of Genes with SNPs	No. of Identified Diseases	Species-Specific Diseases *
Human vs. rhesus macaque	577,417	13,790	1873	2376	53 (42)
Human vs. marmoset	516,545	12,090	1777	2283	24 (20)
Human vs. pig	395,787	9376	1597	2093	23 (18)
Human vs. mouse	264,070	5923	1336	1709	7 (5)
Human vs. rat	256,017	5975	1331	1712	8 (6)

* Numbers inside brackets indicate the number of genes.

## Data Availability

The data presented in this study are openly available in NCBI (National Center for Biotechnology Information) and the data accession numbers were provided in Section 2.

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
