# Peer review of "Evaluation of Five Mammalian Models for Human Disease Research Using Genomic and Bioinformatic Approaches"

_biomedicines, 2023, doi:10.3390/biomedicines11082197_

Round 1

Reviewer 1 Report

The paper is interesting. There are a few questions need to be addressed.

1.     The authors retrieved genomic data for human, rhesus macaque, pig, mouse and rat from Ensembl, why did they retrieve genomic data for marmoset from NCBI (GCF_009663435.1) (Line 72-77) instead of Ensembl (GCA_011100555.1)?  It would be more comparable when using an identical data source.

2.     Is it appropriate/rational to determine tissue-specific relevance among different species by comparing tissue-specific transcriptomic data? Transcriptome varies in different stages of development and environmental conditions compared to the stable genome. 

3.     The authors concluded “In the gastrointestinal tissues (colon, spleen), the pig showed the highest correlation while mouse displayed a better correlation in the heart and kidney” in the abstract (Line 31-33). While in Figure 5, the rat had a higher correlation coefficient with human than pig in the colon (0.3 vs. 0.29). what’s more, the correlation coefficients (<0.3) in most tissues except skeletal muscle were too low to show any correlations, which further indicated that transcriptome was not suitable for this comparison regarding the high percentage of CDS identities among human and the three animal models (86.53%-89.37%). 

Author Response

The paper is interesting. There are a few questions need to be addressed.

Thank you for providing your constructive feedback. We have addressed all the concers. Below, please see our point-by-point reponses.

  1. The authors retrieved genomic data for human, rhesus macaque, pig, mouse and rat from Ensembl, why did they retrieve genomic data for marmoset from NCBI (GCF_009663435.1) (Line 72-77) instead of Ensembl (GCA_011100555.1)? It would be more comparable when using an identical data source.

Complete genome sequence for the Marmoset was unavailable at Ensembl as of 2021, when this analysis was carried out. As it is available now, we have downloaded the Marmoset CDS from the ENSEMBL and mapped the CDS against the NCBI. We have provided the mapped ENSEMBL id and gene name in the Supplementary Table S3 (tab Human vs. Marmoset).

  1. Is it appropriate/rational to determine tissue-specific relevance among different species by comparing tissue-specific transcriptomic data? Transcriptome varies in different stages of development and environmental conditions compared to the stable genome.

We agree with the reviewer that different conditions change gene expression differently. While this caveat exists, our intention on performing the transcriptome analysis is to identify any similarity in gene expression between species using the Gene Expression Atlas database. Analysis like this may provide an overview of expression similarities between species because the comparisons are made between tissue-specific datasets from each species. We have stated the limitation of this analysis in the discussion section (lines 368-372) as follows: “Comparing the tissue expression data between the species has limitations, mainly due to the difference in developmental stages and environmental conditions of the subjects at the time of tissue procurement and also the differences in the quality of the data. However, this analysis provides an overview of the similarities and dissimilarities of the tissue spe-cific expression between the species.”

  1. The authors concluded “In the gastrointestinal tissues (colon, spleen), the pig showed the highest correlation while mouse displayed a better correlation in the heart and kidney” in the abstract (Line 31-33). While in Figure 5, the rat had a higher correlation coefficient with human than pig in the colon (0.3 vs. 0.29). what’s more, the correlation coefficients (<0.3) in most tissues except skeletal muscle were too low to show any correlations, which further indicated that transcriptome was not suitable for this comparison regarding the high percentage of CDS identities among human and the three animal models (86.53%-89.37%).

We have updated the correlation analysis (Figure 5) and corrected the mistakes in the abstract. We agree with the reviewer that because of the analysis with non-stratified datasets, the correlation coefficient is low although the CDS identities are high. Our effort was to grossly identify similarities, if any, within these datasets. To clarify the drawback of this analysis to the readers, we have further discussed the limitations of this analysis in the discussion section (lines 372-378): “As we have not stratified the expression datasets in terms of age, sex, developmental stage, environmental conditions or other relevant parameters which influences gene expres-sions, the analysis only shows an approximation of the expression correlation between species. This reason might explain the low correlation coefficients that were identified between these species (around 0.3), even though the sequence identities at the CDS level between species are high.”

Reviewer 2 Report

Summary:

              The house mouse is frequently used to study human disease.  However, many differences between house mouse and human, including much smaller body sizes and inability to get some human diseases naturally, make the house mouse sub-optimal for studying many human diseases.  This study presents an approach to identify useful animal models for human disease by identifying animals with orthologs of human protein-coding SNPs associated with diseases and by identifying animals whose gene expression in tissues relevant to diseases is highly correlated with that of humans in that tissue.  The study applies this approach to rhesus macaque, marmoset, house mouse, Norway rat, and pig and recommends different animal models for different types of diseases.

Major comments:

1.  Parts of the methodology were not entirely clear to me, which made interpreting some of the results difficult.  I have pointed out the unclear parts in the Minor comments.  Clarifying these would substantially improve the paper.

2.  My understanding is that methods do not account for linkage disequilibrium.  The idea of linkage disequilibrium is that, if two SNPs are associated with a disease and are close to each other in the genome, recombination events are unlikely to happen between the SNPs, so it is possible that only one SNP’s allele contributes to causing the disease and the other SNP’s alleles’ occurrences are correlated with the causal SNPs alleles’ occurrences due its proximity to the causal SNP in the genome.  This can make quantifications of number of SNPs or number of diseases associated with SNPs misleading.  I recommend either accounting for LD or showing that at most a few SNPs are close to other SNPs.

3.  I am concerned that some of the results regarding mapping SNPs to different species are an artifact of differences in genome assembly quality.  For example, the mouse assembly used is one of the best genome assemblies currently available, so having more SNPs or SNPs associated with more diseases mapping to that assembly than to other assemblies might be a result of its superior quality.  I would recommend either providing some assembly quality information that would support that all assemblies are sufficiently high quality to not cause major differences in the ability to map SNPs across species or explicitly stating that differences in assembly quality may partially explain some of the results.

Minor comments:

Abstract:

1.  The study seems to take two approaches to identify useful model organisms for studying disease: 1) Finding mammals to which protein-coding SNPs map and 2) Finding mammals with similar gene expression to humans in tissues relevant to the disease.  The current version of the abstract reads as if each of these is a separate study.  I recommend re-writing it to make the two approaches sound more like a single, unified story.

Introduction:

1.  In line 45, I would replace “larger animal models” with something like “animal models with more disease-relevant similarities to human” because size is one of multiple reasons that mice are not ideal models for some diseases.

2.  In line 63, the meaning of “common and unique clinically associated SNPs” was a little unclear.  Common could be interpreted to mean SNPs that vary in the human population as well as in the populations of other species in the study, it could be interpreted to mean SNPs where some of the orthologs of the human CDS have the minor allele, or it could be interpreted to mean that the position with the SNP maps to other species.  I recommend clarifying this.

3.  I would add a sentence or two to the end of the Introduction explaining how the two approaches fit together and summarizing the biological results.

Materials and Methods:

1.  I would add an explanation of exactly how the CDS were defined for a given gene in a species.  For example, if the CDS come from a gene with multiple proteins due to alternative splicing, I would either state that all protein-coding exons were used, or, if they were not, explain how protein-coding exons were selected.

2.  I would explain exactly how CDS were queried using BLAST.  For example, were all CDS concatenated, or was each exon queried separately?

3.  In line 94, I would state the exact definition of “conserved sequence.”

4.  In line 102, I would replace “SNPs” with “human SNPs” if only human SNPs were used.

5.  My understanding of the sentence in lines 102-106 is that the SNPs were used to predict the disease of the human in the reference genome.  I think that this understanding is probably incorrect.  I recommend clarifying this sentence.

6.  In lines 106-109, it would be helpful if the authors could define “conserved” and “species-specific” SNPs.

7.  It would be helpful to add a supplemental table with the exact gene expression datasets that were downloaded as well as the citations for the papers that those datasets came from.

8.  Pearson correlation can sometimes be driven by outliers, so I would recommend also using Spearman correlation for comparing gene expression in a tissue between species.

Results:

1.  In lines 174-176, I would state that the chromosome coloring is for visualization purposes.

2.  In line 220, defining “predicted SNPs” would be helpful.

3.  In lines 220-222, I was surprised that fewer SNPs map to primates than to other species since the other species are much more distantly related to humans than primates are.  It would be helpful to add an explanation of why this might happen.  If my understanding of SNP mapping is incorrect, it would be helpful to clarify what is meant by mapping SNPs.

4.  I recommend investigating the quality of the RNA-seq data and adding sub-section to the Results or Methods describing the quality information and either why data quality differences are unlikely to explain the results or what caveats the results might have due to data quality differences.

5.  I recommend adding a paragraph to the Results section comparing the results from the SNP mapping analysis to the results from the gene expression analysis.

Discussion:

1.  The sentence in lines 343-345 suggests that a whole-genome comparison between human and pig as well as a cancer comparison between human and pig were done in this study.  I would revise the sentence to either limit it to only the investigations from this study or include citations from relevant investigations done in previous studies.

2.  I would add a paragraph describing this study’s limitations to the Discussion.  I have listed some of them in my major comments and other minor comments.  Another is that this study focusses on only GWAS SNPs in CDS, and the majority of GWAS SNPs are elsewhere in the genome.

Figures:

1.  In the Figure 1 caption, I would explain how the pairs of species used for the statistical tests were selected.

2.  Were the p-values in Figure 1 corrected for multiple hypothesis testing?  If so, I would add what correction was done.  If not, I would do a multiple hypothesis correction, revise the p-values if necessary, and add the correction to the caption.

3.  In Figure 2, I might change the color-coding scheme so that corresponding chromosomes across species are the same color.  For example, if human chromosome 3 corresponds to macaque chromosome 2, I might make those two chromosomes the same color.  Should this change be implemented, I would also add a supplemental table indicating how the chromosomes were mapped.

4.  Since Pearson correlations can be driven by outliers, I would replace the Pearson correlations with Spearman correlations and make the Pearson correlations a supplementary figure.  I would also add supplementary tables with p-values for the Spearman and Pearson correlations.

Supplementary Tables:

1.  In Supplementary Tables S1 and S3, I would add an explanation in the caption of what the percentages are.

2.  In Supplementary Table S2, I would make the “non coding genes” row labels a little more descriptive.

Supplementary Figures:

1.  I would add a description of the color scheme to Supplementary Figure S1.

2.  Supplementary Figures S2-S6 have huge numbers of panels, none of which are especially informative.  I recommend removing them.

3.  Supplementary Figures S7-S12 are so large that they are hard to read.  I recommend removing the gene names, making the figures shorter, and then adding a supplemental table with the gene names and corresponding expression values.

A few terms were not clearly defined, and they have been pointed out in my Minor Comments.  Other than that, the paper was coherently written.

Author Response

Summary:

              The house mouse is frequently used to study human disease.  However, many differences between house mouse and human, including much smaller body sizes and inability to get some human diseases naturally, make the house mouse sub-optimal for studying many human diseases.  This study presents an approach to identify useful animal models for human disease by identifying animals with orthologs of human protein-coding SNPs associated with diseases and by identifying animals whose gene expression in tissues relevant to diseases is highly correlated with that of humans in that tissue.  The study applies this approach to rhesus macaque, marmoset, house mouse, Norway rat, and pig and recommends different animal models for different types of diseases.

Thank you for providing your constructive feedback. We have addressed all the concers. Below, please see our point-by-point reponses.

Major comments:

  1. Parts of the methodology were not entirely clear to me, which made interpreting some of the results difficult. I have pointed out the unclear parts in the Minor comments.  Clarifying these would substantially improve the paper.

We thank the reviewer for helping us resolve these issues. We have addressed all the comments and provided a point-by-point response below.

  1. My understanding is that methods do not account for linkage disequilibrium. The idea of linkage disequilibrium is that, if two SNPs are associated with a disease and are close to each other in the genome, recombination events are unlikely to happen between the SNPs, so it is possible that only one SNP’s allele contributes to causing the disease and the other SNP’s alleles’ occurrences are correlated with the causal SNPs alleles’ occurrences due its proximity to the causal SNP in the genome.  This can make quantifications of number of SNPs or number of diseases associated with SNPs misleading.  I recommend either accounting for LD or showing that at most a few SNPs are close to other SNPs.

We understand the reviewer’s concern and agree that this study does not account for linkage disequilibrium (LD). In this study, we are focusing only on the conserved protein coding regions (CDS) of the genome across the species and SNP mapping was done based on the conservation of the major allele SNP position between the human and other species. A whole genome sequence-based approach would help carry out such analysis, which is not within the scope of this project.

  1. I am concerned that some of the results regarding mapping SNPs to different species are an artifact of differences in genome assembly quality. For example, the mouse assembly used is one of the best genome assemblies currently available, so having more SNPs or SNPs associated with more diseases mapping to that assembly than to other assemblies might be a result of its superior quality.  I would recommend either providing some assembly quality information that would support that all assemblies are sufficiently high quality to not cause major differences in the ability to map SNPs across species or explicitly stating that differences in assembly quality may partially explain some of the results.

We thank you reviewer for pointing out this issue. We have provided the assembly quality information in the supplemental information (Supplementary Table S1), and added a sentence in the discussion regarding this limitation.The download genomes are assembled at the chromosomal level with the scaffold N50 from ~14 to 106 million base pairs, which showed good assembly quality; we have retrieved the CDS sequence for our analysis. The genome assembly information such as genome length, size, scaffold and contig N50 and L50 and assembly level were provided in Supplementary Table S1.

Minor comments:

Abstract:

  1. The study seems to take two approaches to identify useful model organisms for studying disease: 1) Finding mammals to which protein-coding SNPs map and 2) Finding mammals with similar gene expression to humans in tissues relevant to the disease. The current version of the abstract reads as if each of these is a separate study.  I recommend re-writing it to make the two approaches sound more like a single, unified story.

 We have modified the abstract to portray the unified story of this study (line 22).

Introduction:

  1. In line 45, I would replace “larger animal models” with something like “animal models with more disease-relevant similarities to human” because size is one of multiple reasons that mice are not ideal models for some diseases.

We have changed the wording according to the suggestion on line number 45.

  1. In line 63, the meaning of “common and unique clinically associated SNPs” was a little unclear. Common could be interpreted to mean SNPs that vary in the human population as well as in the populations of other species in the study, it could be interpreted to mean SNPs where some of the orthologs of the human CDS have the minor allele, or it could be interpreted to mean that the position with the SNP maps to other species.  I recommend clarifying this.

We have clarified the meaning of “common” in line 63. “SNPs that are present in human and predicted to be present in other species”.

  1. I would add a sentence or two to the end of the Introduction explaining how the two approaches fit together and summarizing the biological results.

 We have provided a sentence at the end to summarize the study (lines 68-71). “Taken together, the genome and transcriptome comparison performed in this study will provide some insights into selecting suitable animal models to study human diseases.”

Materials and Methods:

  1. I would add an explanation of exactly how the CDS were defined for a given gene in a species. For example, if the CDS come from a gene with multiple proteins due to alternative splicing, I would either state that all protein-coding exons were used, or, if they were not, explain how protein-coding exons were selected.

A CDS is a DNA sequence that represents all the protein-coding exons concatenated into one continuous sequence. This is now explained in the manuscript at lines 85 and 86.

  1. I would explain exactly how CDS were queried using BLAST. For example, were all CDS concatenated, or was each exon queried separately?

All CDS from human and non-human species were downloaded from the Ensembl database except the marmoset, which was obtained from NCBI.  Each species-specific set of CDS was queried against the human set of CDS using blastn. Each non-human CDS was queried independently (not as concatenated) against the human CDS.  Querying was done at each CDS level, not at the exon-level. We have made corresponding changes in the main manuscript at line no: 90-96

  1. In line 94, I would state the exact definition of “conserved sequence.”

We have defined the parameters to identify conserved sequence in lines 95-99 as follows. “Based on the pairwise alignment, we identified conserved CDS in other mammalian models against the human, where a conserved sequence is defined as a single contiguous sequence from each species that also pass the following filters. To avoid the false positives, we selected only those sequences with greater than 50% identity and covered at least 50% length of the human CDS.”

  1. In line 102, I would replace “SNPs” with “human SNPs” if only human SNPs were used.

We have used human SNPs and have added the word “human” in the line number 109.

  1. My understanding of the sentence in lines 102-106 is that the SNPs were used to predict the disease of the human in the reference genome. I think that this understanding is probably incorrect.  I recommend clarifying this sentence.

The SNPs were searched to identify association with human diseases. We have modified the sentence to clarify the process in line number 111 to 113

  1. In lines 106-109, it would be helpful if the authors could define “conserved” and “species-specific” SNPs.

We have added clarification of these two terms in the manuscript (line 116 to 118).

  1. It would be helpful to add a supplemental table with the exact gene expression datasets that were downloaded as well as the citations for the papers that those datasets came from.

We have added the specific datasets that were used in the study in lines 131-134 with citations.

  1. Pearson correlation can sometimes be driven by outliers, so I would recommend also using Spearman correlation for comparing gene expression in a tissue between species.

We have updated Figure 5 with Spearman correlation.

Results:

  1. In lines 174-176, I would state that the chromosome coloring is for visualization purposes.

We have updated the sentence on line number 186.

  1. In line 220, defining “predicted SNPs” would be helpful.

We have provided clarification of the predicted SNPs on line number 231 and 232.

  1. In lines 220-222, I was surprised that fewer SNPs map to primates than to other species since the other species are much more distantly related to humans than primates are. It would be helpful to add an explanation of why this might happen.  If my understanding of SNP mapping is incorrect, it would be helpful to clarify what is meant by mapping SNPs.

Thank you for pointing this out. We have repeated our analysis to address this issue, and fixed the problem. In the previous version of the manuscript, our program only counted the minor alleles but not the major alleles, which resulted in the observed surprising trend. We have corrected this mistake in our code and repeated the analysis only for the major allele counts for human SNPs that are observed in the animal models. The new anlayis gave us the expected trends, where the closely related species have more number of conserved SNPs compared to the distantly related. Based on these results, we also reanalyzed the downstream tables and figures and modified the results and discussion sections, accordingly. Please find tracked changes in the main manuscript. Tables 3 and 4; figures 3 and 4; and supplementary Tables S7, S8 and S9 are modified based on the new analysis.

  1. I recommend investigating the quality of the RNA-seq data and adding sub-section to the Results or Methods describing the quality information and either why data quality differences are unlikely to explain the results or what caveats the results might have due to data quality differences.

We used the RNA-seq data with a default minimum expression value of 0.5 Transcripts Per Million (TPM). We have addressed the caveats based on the other reviewer’s comment and an explanation was provided in the discussion at lines 368-372.

  1. I recommend adding a paragraph to the Results section comparing the results from the SNP mapping analysis to the results from the gene expression analysis.

Because the expression data is not disease-specific and only represents an overall tissue-specific correlation between species, we are unable to connect the SNP association to diseases with the expression data. We agree that it would be beneficial for the study if disease-specific expression data is available.

Discussion:

  1. The sentence in lines 343-345 suggests that a whole-genome comparison between human and pig as well as a cancer comparison between human and pig were done in this study. I would revise the sentence to either limit it to only the investigations from this study or include citations from relevant investigations done in previous studies.

We have added a sentence describing that the conclusions of this study are dependent on the selection of the datasets on line 348 to 351.

  1. I would add a paragraph describing this study’s limitations to the Discussion. I have listed some of them in my major comments and other minor comments.  Another is that this study focusses on only GWAS SNPs in CDS, and the majority of GWAS SNPs are elsewhere in the genome.

We have added sentences about limitations of this study in response to the reviewers’ comments and in discussion on line 368 to 378.

Figures:

  1. In the Figure 1 caption, I would explain how the pairs of species used for the statistical tests were selected.

We added the details in the figure legend on lines 175.

  1. Were the p-values in Figure 1 corrected for multiple hypothesis testing? If so, I would add what correction was done.  If not, I would do a multiple hypothesis correction, revise the p-values if necessary, and add the correction to the caption.

We have used Dunn’s multiple comparison test and added the information in the figure legend.

  1. In Figure 2, I might change the color-coding scheme so that corresponding chromosomes across species are the same color. For example, if human chromosome 3 corresponds to macaque chromosome 2, I might make those two chromosomes the same color.  Should this change be implemented, I would also add a supplemental table indicating how the chromosomes were mapped.

We have used a different color scheme, where the chromosomes are similarly color coded across the species. This method helps to visualize which chromosome from other species shows similarity with a given human chromosome.

  1. Since Pearson correlations can be driven by outliers, I would replace the Pearson correlations with Spearman correlations and make the Pearson correlations a supplementary figure. I would also add supplementary tables with p-values for the Spearman and Pearson correlations.

We have replaced the figure with Spearman correlation and provided a supplementary table (S10) with P-values from both comparisons.

Supplementary Tables:

  1. In Supplementary Tables S1 and S3, I would add an explanation in the caption of what the percentages are.

We have provided the explanation in the caption.

  1. In Supplementary Table S2, I would make the “non coding genes” row labels a little more descriptive.

We have provided the description

Non coding genes (genes that are not expressed, e.g., transfer RNA, ribosomal RNA, etc.).

Supplementary Figures:

  1. I would add a description of the color scheme to Supplementary Figure S1.

We provided the color scheme as follows.

Figure S1, title: The number of disease-associated SNPs were plotted in a circus plot. The coloured circle differentiated the six different organisms (outer –> inner: human, rhesus macaque, marmoset, pig, mouse, and rat), and the red line inside each circle represents the disease-associated SNPs;

  1. Supplementary Figures S2-S6 have huge numbers of panels, none of which are especially informative. I recommend removing them.

Based on reviewer’s suggestion, the Supplementary Figures S2-S6 were removed.

  1. Supplementary Figures S7-S12 are so large that they are hard to read. I recommend removing the gene names, making the figures shorter, and then adding a supplemental table with the gene names and corresponding expression values.

 We agree with the reviewer that these data are large and hard to read, the Supplementary Figures S7-S12 were removed.

Comments on the Quality of English Language

A few terms were not clearly defined, and they have been pointed out in my Minor Comments.  Other than that, the paper was coherently written.

Thank you for the comment. We also had the paper read by a native speaker to improve the readability.

Round 2

Reviewer 2 Report

Summary:

              The authors have substantially improved the manuscript by adding supplementary tables providing necessary details to interpret the results, clarifying definitions of terminology, and being more transparent about the study’s limitations.

Major comments:

1.  The Results section describes some “model-specific” diseases, which seem to be diseases whose associated SNPs map to exactly one other species.  I am not sure if these diseases are truly model-specific or if this result is an artifact of the relevant GWAS containing a much smaller number of SNPs than other GWAS and, if the GWAS had more SNPs, there would be SNPs mapping to multiple species.  One way to investigate this issue would be to take a very large GWAS that has SNPs mapping to multiple species, randomly down-sample SNPs, determine the number of down-sampled SNPs that lead to having SNPs mapping to only one species, and show that this number is smaller than the number of SNPs in the GWAS for most “model-specific” diseases.  Alternatively, the authors could add the number of SNPs in the GWAS for each disease and the number of SNPs that map to the other species as columns in Supplementary Table S9 so that readers can consider this when interpreting the results.

2.  The other reviewer mentioned that gene expression results may be more explained by the age of the animal that the data came from or the environment in which the animal lived than the species of the animal, which is a concern that I share.  In addition, the results might be dependent on the sex of the animal and the amount of time that the animal had been awake when it was dissected, both of which have been shown to play major roles on gene expression.  While I appreciate the authors’ transparency about the existence of these limitations, determining the extent of these limitations’ effects on the results would require a substantial amount of work for readers.  To facilitate readers’ interpretations of the gene expression results, the authors should add a supplemental table with the age, sex, time since waking up if that is available, disease status, and whether animals were raised in captivity or in the wild for all animals whose gene expression data were analyzed in this paper so that readers can consider this information when interpreting the results.  Alternatively, the authors could remove the gene expression section, as it is not a large part of the manuscript, and there is little analysis connecting the results in it to the other results in the paper.

3.  The gene expression results might also be dependent on data quality.  For example, the Abstract has a new sentence that the mouse’s gene expression had the highest correlation with human gene expression, but RNA-seq has been done many more times in mouse than in other mammals, so RNA-seq protocols may work better in mouse, leading to higher data quality that may partially explain this result.  I appreciate that the authors now describe how they filtered the data; if they keep the gene expression section, it would be great if they could also provide in the supplement results from any quality control analysis they did as well as a summary of the results of quality control analysis done in the manuscripts where the data was published so that readers can consider this when interpreting the results.

Minor comments:

Introduction:

1.  The idea of mapping SNPs to other species is still a little unclear in the Introduction.  This could mean finding SNPs where the position is mappable to the other species, even if the nucleotide changes.  It could mean finding SNPs for which the corresponding nucleotide in the other species is the major allele, which the Methods suggest is the correct definition.  It could mean finding SNPs for which the corresponding nucleotide in the other species is the non-disease allele.  It could mean finding SNPs for which the corresponding nucleotide in other species also varies across individuals within that species.  Since this is crucial to understanding the questions addressed in this paper, I would add a sentence to the Introduction defining this explicitly.

Materials and Methods:

1.  The major allele is often the non-disease allele, but it is not guaranteed to be the non-disease allele.  I think that mapping the non-disease allele might be better for understanding conservation of the healthy state than mapping the major allele.  Since these alleles are usually the same, the authors could, alternatively, report how often the major allele is the non-disease allele and, if this does not always occur, mention that the major allele occasionally being the disease allele is a limitation of the study.

2.  I was a little confused by what the authors meant when they said that they included only genes with a minimum expression of 0.5 TPM.  Does this mean that genes needed to have this minimum expression in all species to be included?  Does it mean that, if a gene had a lower expression in some species, that gene, species combination was excluded?  Does this mean that genes with expressions below 0.5 TPM had their expressions set to 0 for the analyses?  It would be helpful if the authors clarified this.

3.  Since the authors have now added Spearman correlation analyses, I would change “Pearson” in line 205 to “Pearson and Spearman.”

Results:

1.  In line 325, I would replace “identified” with something like “mapped to a nucleotide that was the same as the human major allele.”

2.  I appreciate how the authors have corrected the numbers of SNPs that map to different species.  I am still a little surprised that more SNPs map to pig than to mouse and rat given that mouse and rat diverged from human more recently than pig did.  This might be because mice and rats have shorter generation times than pigs.  I think that the authors also cited some papers in the Introduction that support this finding.  It would be helpful if the authors added some potential explanations of this finding to the Results or Discussion sections.

3.  The Results mention that no GWAS SNPs that map to other species were identified on chromosome Y.  I am not sure if this is a result of chromosome Y being less conserved, diseases on chromosome Y being human-specific, or SNPs on chromosome Y not being included in many GWAS.  It would be helpful if the authors could provide some potential explanations for this finding in either the Results or Discussion section.

4.  I would delete the sentence in lines 532-533 because it is redundant with an earlier sentence.

Discussion:

1.  In line 652, I would change “reproductive system diseases” to “reproductive system and metabolic diseases.”

2.  If there is any literature about the roles of PTCH1 and STK11 in pigs that is relevant to cancer, the authors could strengthen the paper by citing it and including a couple sentences summarizing it.

3.  If the authors are keeping the gene expression section, I would add a few sentences comparing the results in it to the results from the GWAS SNP mapping.

Figures:

1.  The species names in Figures 1b and 3a are a little hard to read.  I would make them larger.

2.  I appreciate that the authors have now clarified that they used Dunn’s multiple comparison test to determine which species has a shifted upward percent identity.  Was a Bonferroni or other adjustment used?  If so, the authors should specify what adjustment was used.  If not, the authors should run a p-value adjustment method since they are presenting results from multiple tests and use those p-values.  Adding a supplementary table with the Kruskal-Wallis p-values and Dunn multiple comparison test results that went into Figure 1 would also be helpful.

Tables:

1.  I would move Table 4 to the supplement, as I do not think that it makes an essential contribution to the story in the paper.

Supplementary Tables:

1.  Supplementary Table S1 is an extremely helpful addition.  I recommend defining “Small non-coding genes” and “Long non-coding genes” in the caption.

2.  In Supplementary Tables S7-8, I would add the genome assembly corresponding to the coordinates on each sheet to the sheet header or the caption.

I found a number of grammar mistakes as I was reading the paper.  They were especially prevelant in the sentences that had been added since the previous submission.  I recommend carefully proof-reading the manuscript before re-submitting.

Author Response

Comments and Suggestions for Authors

Summary:
The authors have substantially improved the manuscript by adding supplementary tables providing necessary details to interpret the results, clarifying definitions of terminology, and being more transparent about the study's limitations.
Thank you for helping to improve the quality of our manuscript.

Major comments:
1. The Results section describes some "model-specific" diseases, which seem to be diseases whose associated SNPs map to exactly one other species. I am not sure if these diseases are truly modelspecific or if this result is an artifact of the relevant GWAS containing a much smaller number of SNPs than other GWAS and, if the GWAS had more SNPs, there would be SNPs mapping to multiple species. One way to investigate this issue would be to take a very large GWAS that has SNPs mapping to multiple species, randomly down-sample SNPs, determine the number of downsampled SNPs that lead to having SNPs mapping to only one species, and show that this number is smaller than the number of SNPs in the GWAS for most "model-specific" diseases.
Alternatively, the authors could add the number of SNPs in the GWAS for each disease and the number of SNPs that map to the other species as columns in Supplementary Table S9 so that readers can consider this when interpreting the results.

Response: We understand the reviewer’s concerns regarding potential biasedness based on the size of GWAS studies; however, this study did not use any specific GWAS study. Instead, the study used only the curated and annotated SNPs containing a reference SNP (rs) number from dbSNP. dbSNP database is a gold standard (with more than 20 years of history) and composed of over 2 billion submitted variants from thousands of studies, which are regularly clustered, integrated and annotated. Because we didn’t select a specific GWAS study for SNP mapping in this project, we believe that the reviewer’s concerns are addressed.
As per the reviewer's suggestion, the number of SNPs in the GWAS for each disease and the number of SNPs that map to the other species were listed in Supplementary Table S10 (GWAS_for_each_disease), which is the new number in this revision.

2. The other reviewer mentioned that gene expression results may be more explained by the age of the animal that the data came from or the environment in which the animal lived than the species of the animal, which is a concern that I share. In addition, the results might be dependent on the sex of the animal and the amount of time that the animal had been awake when it was dissected, both of which have been shown to play major roles on gene expression. While I appreciate the authors' transparency about the existence of these limitations, determining the extent of these limitations' effects on the results would require a substantial amount of work for readers. To facilitate readers' interpretations of the gene expression results, the authors should add a supplemental table with the age, sex, time since waking up if that is available, disease status, and whether animals were raised in captivity or in the wild for all animals whose gene expression data were analyzed in this paper so that readers can consider this information when interpreting the results. Alternatively, the authors could remove the gene expression section, as it is not a large part of the manuscript, and there is little analysis connecting the results in it to the other results in the paper.

Response: We agree that the gene expression is not a suitable indicator of the similarity among species given its dynamic nature and sensitivity to the external stimuli and other factors. As suggested by the reviewer, we have competely removed the gene expression data and corresponding figures/tables from the study.

3. The gene expression results might also be dependent on data quality. For example, the Abstract has a new sentence that the mouse's gene expression had the highest correlation with human gene expression, but RNA-seq has been done many more times in mouse than in other mammals, so RNA-seq protocols may work better in mouse, leading to higher data quality that may partially explain this result. I appreciate that the authors now describe how they filtered the data; if they keep the gene expression section, it would be great if they could also provide in the supplement results from any quality control analysis they did as well as a summary of the results of quality control analysis done in the manuscripts where the data was published so that readers can consider this when interpreting the results.

Response: We totally understand these limitations and removed the gene expression data as suggested in point #2.

Minor comments:
Introduction:
1. The idea of mapping SNPs to other species is still a little unclear in the Introduction. This could mean finding SNPs where the position is mappable to the other species, even if the nucleotide changes. It could mean finding SNPs for which the corresponding nucleotide in the other species is the major allele, which the Methods suggest is the correct definition. It could mean finding SNPs for which the corresponding nucleotide in the other species is the non-disease allele. It could mean finding SNPs for which the corresponding nucleotide in other species also varies across individuals within that species. Since this is crucial to understanding the questions addressed in this paper, I would add a sentence to the Introduction defining this explicitly.
Response: We have revised the Introduction as the following to clarify the process (lines 78-84) “Multiple sequence alignment was performed across the six species using conserved CDS and mapped positions in each species corresponding to the human single nucleotide polymorphisms (SNPs) were extracted from the alignment. The mapped SNPs were queried for disease associations to identify common (human SNPs that are identified in all other species) and species-specific clinically-associated SNPs to better define the relevance of an animal model with various human diseases.”

Materials and Methods:
1. The major allele is often the non-disease allele, but it is not guaranteed to be the non-disease allele. I think that mapping the non-disease allele might be better for understanding conservation of the healthy state than mapping the major allele. Since these alleles are usually the same, the authors could, alternatively, report how often the major allele is the non-disease allele and, if this does not always occur, mention that the major allele occasionally being the disease allele is a limitation of the study.
Response: This is a good point and we concur that the mere presence of the major or a minor allele does not guarantee a disease state due to the differences in their penetrance, which vary for each loci. Because this is a study focused on identifying global similarities among species, delving into how often the major vs minor alleles are disease-associated and their penetrance levels would be topics for future studies. As suggested by the reviewer, we have added the possible limitations of assuming the major allele as the disease allele (lines 147 and 149) as follows.
“It should be noted that the major as well as any minor allele may serve as a disease allele depending on their penetrance levels and other covariates, which are not specifically analyzed in this study.”

2. I was a little confused by what the authors meant when they said that they included only genes with a minimum expression of 0.5 TPM. Does this mean that genes needed to have this minimum expression in all species to be included? Does it mean that, if a gene had a lower expression in some species, that gene, species combination was excluded? Does this mean that genes with expressions below 0.5 TPM had their expressions set to 0 for the analyses? It would be helpful if the authors clarified this.
Response: We have removed the gene expression data sections from methods and results in response to the previous comments.

3. Since the authors have now added Spearman correlation analyses, I would change "Pearson" in line 205 to "Pearson and Spearman."
Response: We have removed those content, which was related to the gene expression

Results:
1. In line 325, I would replace "identified" with something like "mapped to a nucleotide that was the same as the human major allele."
Response: We have changed the wording as the following according to the reviewer's suggestion (lines 311 to 312) "reference SNPs from the dbSNP database with a matching nucleotide to the human major allele
in other genomes "

2. I appreciate how the authors have corrected the numbers of SNPs that map to different species.
I am still a little surprised that more SNPs map to pig than to mouse and rat given that mouse and rat diverged from human more recently than pig did. This might be because mice and rats have shorter generation times than pigs. I think that the authors also cited some papers in the Introduction that support this finding. It would be helpful if the authors added some potential explanations of this finding to the Results or Discussion sections.
Response: We have explained the reason for this observation in the discussion section (lines 416- 420) as the following.
"Although mouse and rat are phylogenetically closer to human than pig [10], possible reasons for having more number of SNPs mapped to pig compared to rodents could be due to (i) the shortened generation time in rodents leading to expedited divergence and (ii) this study covers only a subset (10,316 conserved CDS) of the total genome and variations present in the entire genome were not accounted for here."

3. The Results mention that no GWAS SNPs that map to other species were identified on chromosome Y. I am not sure if this is a result of chromosome Y being less conserved, diseases on chromosome Y being human-specific, or SNPs on chromosome Y not being included in many GWAS. It would be helpful if the authors could provide some potential explanations for this finding in either the Results or Discussion section.
Response: Among the 10,316 conserved CDS, only seven CDS were mapped with the human Y chromosome that contained 48 to 130 SNPs across different species with an rs ID. Howver, none of these SNPs were associated with any human disease. That is why none of the disorders were noted in chromosome Y.

4. I would delete the sentence in lines 532-533 because it is redundant with an earlier sentence.
Response: We removed the part of the sentence that stated the APC as the common gene in all species.

Discussion:
1. In line 652, I would change "reproductive system diseases" to "reproductive system and metabolic diseases."
Response: We have made the change it as suggested on line 436.

2. If there is any literature about the roles of PTCH1 and STK11 in pigs that is relevant to cancer, the authors could strengthen the paper by citing it and including a couple sentences summarizing it.
Response: There is no evidence linking PTCH1 to cancer development in pigs, but we have added a sentence (line 465 to 467) referencing an ongoing work that utilizes STK11 to develop a pig model of lung cancer.
“STK11 was targeted to create a pig model of lung cancer that shows evidence of inflammation; however, more work needs to be done to fully develop the model [38].”

3. If the authors are keeping the gene expression section, I would add a few sentences comparing the results in it to the results from the GWAS SNP mapping.
Response: We have removed the gene expression data in response to previous comments.

Figures:
1. The species names in Figures 1b and 3a are a little hard to read. I would make them larger.
Response: The font size was increased in Fig. 1b and 3a.

2. I appreciate that the authors have now clarified that they used Dunn's multiple comparison test to determine which species has a shifted upward percent identity. Was a Bonferroni or other adjustment used? If so, the authors should specify what adjustment was used. If not, the authors should run a p-value adjustment method since they are presenting results from multiple tests and use those p-values. Adding a supplementary table with the Kruskal-Wallis p-values and Dunn multiple comparison test results that went into Figure 1 would also be helpful.
In this revision, we provided the p-values from Kruskal-Wallis test, Dunn's multiple comparison tests and Bonferroni's multiple comparison tests in the supplementary table S4.

Tables:
1. I would move Table 4 to the supplement, as I do not think that it makes an essential contribution to the story in the paper.
Response: Table 4 was moved to supplementary table S11

Supplementary Tables:
1. Supplementary Table S1 is an extremely helpful addition. I recommend defining "Small noncoding genes" and "Long non-coding genes" in the caption.
Response: We have provided the caption in Supplementary Table S1 "Small non-coding genes - small non-coding genes with size < 200nt (e.g. miRNA, piRNA)
Long non-coding genes (that produce transcripts of over >=200 nucleotides in length which do not appear to be translated) (e.g. lincRNA)."

2. In Supplementary Tables S7-8, I would add the genome assembly corresponding to the coordinates on each sheet to the sheet header or the caption.
Response: The genome assembly used for the analysis is provided in the materials and method section 2.1 and as per suggestion, we have added the assembly information on sheet header of Supplementary Tables S9-S10
Comments on the Quality of English Language I found a number of grammar mistakes as I was reading the paper. They were especially prevalent in the sentences that had been added since the previous submission. I recommend carefully proofreading the manuscript before re-submitting.
Response: Thank you for the suggestion. We got the manuscript edited by a native English speaker to fix grammatical errors. 

Round 3

Reviewer 2 Report

Summary:

              The authors have further improved the manuscript by removing the gene expression section, adding additional wording to clarify the crucial sections for understanding the paper, and fixing many of the grammar mistakes in previous versions.

Major comments:

1.  The authors do not seem to evaluate the statistical significance of the mapping of SNPs associated with disease classes to different species even though they seem to refer to such analyses in multiple parts of the paper (see Minor comments).  If the authors did this kind of evaluation, they should add a table with the results, a paragraph in the Results section summarizing it, and a paragraph in the Materials and Methods describing how they did it.  If they did not do it, one way to do this is to do a hyper-geometric test for each species, disease class combination followed by a Bonferroni correction, though I am not sure if this is the best approach.  Adding a table with these results would make the manuscript much more compelling.  Alternatively, the authors could discuss why there is not a great way to evaluate the statistical significance of these results as a limitation of the study in the Discussion section.

Minor comments:

Materials and Methods:

1.  The method used for mapping SNPs is now completely clear to me.  I appreciate how the authors have added phrasing in multiple parts of the manuscript to help clarify this.  However, I am concerned that this approach might still be confusing to researchers reading this manuscript for the first time.  I recommend adding a figure to illustrate how this process works, including an example of a species-specific mapping.

2.  I think that Supplementary Table S4 might be confusing to readers who have not run Dunn’s multiple comparison tests.  I recommend adding a few sentences to the Materials and Methods about exactly how the test was run and the values in the table were generated.

Results:

1.  The phrase “enriched in pigs” suggests that there were more SNPs associated with cancer, gastrointestinal diseases, and organismal injury and abnormalities than expected by chance.  If this is the case, then the authors should add a p-value and a sentence explaining how statistical significance was evaluated.  If statistical significance was not evaluated, the authors should either evaluate it or re-phrase this to describe the evaluation they did.

Discussion:

1.  The authors state that “the difference in cancer mapping among species was not statistically significant,” but I cannot figure out where the analysis of the lack of significance in the cancer mapping difference between species was done.  If they did this analysis, they should clarify the relevant section.  If they did not do this analysis, they should either add it or re-phrase this part of the Discussion to summarize their analysis more accurately.

Supplementary Tables:

1.  Supplementary Table S4 is a great addition to the manuscript.  I found the rows under “Kruskal-Wallis test” a little confusing.  I recommend revising the headers, explaining them in more detail in the ReadMe, explaining them in more detail in the caption, or explaining them in more detail in the relevant part of the Materials and Methods section (see Materials and Methods Minor comments).

Most of the grammar mistakes in previous versions have been fixed.  I did notice a few minor grammar errors, so I would recommend doing a final proof-read before re-submitting.

Author Response

Point-by-point response to Reviewer’s comments

Reviewer-2

Summary:

The authors have further improved the manuscript by removing the gene expression section, adding additional wording to clarify the crucial sections for understanding the paper, and fixing many of the grammar mistakes in previous versions.

Thank you for helping to improve the quality of our manuscript.

Below we provide point-by-point response to address all the comments by the reviewer. We highlighted the areas where changes were made in Yellow color. We also got the English Editing done by MDPI-approved editors and those changes are shown in tracking throughout the manuscript.

Major comments:

  1. The authors do not seem to evaluate the statistical significance of the mapping of SNPs associated with disease classes to different species even though they seem to refer to such analyses in multiple parts of the paper (see Minor comments). If the authors did this kind of evaluation, they should add a table with the results, a paragraph in the Results section summarizing it, and a paragraph in the Materials and Methods describing how they did it.  If they did not do it, one way to do this is to do a hyper-geometric test for each species, disease class combination followed by a Bonferroni correction, though I am not sure if this is the best approach.  Adding a table with these results would make the manuscript much more compelling.  Alternatively, the authors could discuss why there is not a great way to evaluate the statistical significance of these results as a limitation of the study in the Discussion section.

The reviewer was correct. We have not evaluated the statistical significance of the mapped SNPs with disease classes in different species because we do not have a distribution or sufficient n (number of species) to carry out the statistical significance. Hence, we determined the disease association of SNPs (Figure 4B) for each species as the relative percentage of “number of diseases in each category divided by the total number of diseases associated with all SNPs in that species”. The relative percentage of diseases in each category in each species provides the basis for a comparison across the species. The higher the relative percentage of a disease category, the higher the relevance of that species to study corresponding genetic diseses.

As suggested by the reviewer, we clarified this in the methods section (lines 331 to 334) and also added a sentence in the discussion about this limitation (lines 728 to 732)

Minor comments:

Materials and Methods:

  1. The method used for mapping SNPs is now completely clear to me. I appreciate how the authors have added phrasing in multiple parts of the manuscript to help clarify this.  However, I am concerned that this approach might still be confusing to researchers reading this manuscript for the first time.  I recommend adding a figure to illustrate how this process works, including an example of a species-specific mapping.

As suggested, we have added an illustration as Figure 1 to depict the workflow of the process with an example of SNP mapping in Rhesus macaque.

  1. I think that Supplementary Table S4 might be confusing to readers who have not run Dunn’s multiple comparison tests. I recommend adding a few sentences to the Materials and Methods about exactly how the test was run and the values in the table were generated.

We performed one-way ANOVA and Bonferroni’s multiple comparisons tests using GraphPad Prism 10 software. This information was added in the Methods (lines 316 to 319) and in the Supplementary Table S4 caption sections of each sheet.

Results:

  1. The phrase “enriched in pigs” suggests that there were more SNPs associated with cancer, gastrointestinal diseases, and organismal injury and abnormalities than expected by chance. If this is the case, then the authors should add a p-value and a sentence explaining how statistical significance was evaluated.  If statistical significance was not evaluated, the authors should either evaluate it or re-phrase this to describe the evaluation they did.

As explained above, we were not able to determine the statistical significance, and our observation is solely based on the relative percentage of diseases in each category in each species.  Hence, the use of phrase ‘enriched in pigs’ is a wrong choice of wording on our side.  We corrected this phrase by changing it to ‘relative percentage’ in that sentence (lines 631-632).

Discussion:

  1. The authors state that “the difference in cancer mapping among species was not statistically significant,” but I cannot figure out where the analysis of the lack of significance in the cancer mapping difference between species was done. If they did this analysis, they should clarify the relevant section.  If they did not do this analysis, they should either add it or re-phrase this part of the Discussion to summarize their analysis more accurately.

--It was stated based on the observed differences in the relative percentage of diseases in the cancer category, but we totally agree with the reviewer that it’s not appropriate to mention about statististical significance without any statistical analysis. As suggested, we re-phrased the sentence as follows (lines 728 to 732):

“We mapped the SNP-associated diseases onto different disease groups and observed that the relative frequency of diseases in the cancer category was the highest in the pig; howev-er, we were not able to test these differences statistically because SNP mapping had been carried out against a single reference genome for each species, resulting in a lack of a dis-tribution of values for performing statistical tests”

Supplementary Tables:

  1. Supplementary Table S4 is a great addition to the manuscript. I found the rows under “Kruskal-Wallis test” a little confusing.  I recommend revising the headers, explaining them in more detail in the ReadMe, explaining them in more detail in the caption, or explaining them in more detail in the relevant part of the Materials and Methods section (see Materials and Methods Minor comments).

We have removed the Kruskal-Wallis test and Dunn’s multiple corrections from manuscript, tables and figures, instead results from ANOVA and Bonferroni’s multiple corrections are provided in Supplementary Table S4 and Materials and method lines 428 to 430.

 “Comparisons of the percent identities between the species are statistically significant unless noted by ns = statistically non-significant, as determined via a one-way ANOVA and Bonferro-ni’s multiple comparison test (p-values are provided in Supplementary Table S4).”

Comments on the Quality of English Language

Most of the grammar mistakes in previous versions have been fixed.  I did notice a few minor grammar errors, so I would recommend doing a final proof-read before re-submitting.

As suggested, we have used the editorial services from MDPI for language editing. Please note extensive editing by English Editors throughout the manuscript.